# Multi-scale Whole Slide Image Assessment Improves Deep Learning based WHO 2021 Glioma Classification

**Shubham Innani**                                            SINNANI@IU.EDU
*Department of Pathology and Laboratory Medicine, Indiana University School of Medicine, Indianapolis, IN, USA*

**MacLean P. Nasrallah**              MACLEAN.NASRALLAH@PENNMEDICINE.UPENN.EDU
*Department of Pathology and Laboratory Medicine, Perelman School of Medicine at the University of Pennsylvania, Philadelphia, PA, USA*

**W. Robert Bell**                                              RB27@IU.EDU
*Department of Pathology and Laboratory Medicine, Indiana University School of Medicine, Indianapolis, IN, USA*

**Bhakti Baheti***                                           BVBAHETI@IU.EDU
*Department of Pathology and Laboratory Medicine, Indiana University School of Medicine, Indianapolis, IN, USA*

**Spyridon Bakas***                                          SPBAKAS@IU.EDU
*Department of Pathology and Laboratory Medicine, Indiana University School of Medicine, Indianapolis, IN, USA*
*Equally contributing senior authors*

**Editor:**

## Abstract

The 2021 WHO classification of tumors of the central nervous system necessitates the integration of molecular and histologic profiling for a conclusive diagnosis of glioma. Molecular profiling is time-consuming and may not always be available. We hypothesize that subvisual cues in whole slide images (WSI), not perceivable by the naked eye, carry a predictive value of molecular characteristics and can allow categorization of the adult infiltrative gliomas in one of three major types: i) oligodendroglioma, ii) astrocytoma, and iii) glioblastoma. Towards this end, we present a computational pipeline comprising patch analysis of Hematoxylin and Eosin (H&E)-stained WSIs, feature encoding with ImageNet pretrained ResNet50, and an attention-based multiple instance learning paradigm. We trained individual models at four distinct magnification levels (20x, 10x, 5x, 2.5x), and assessed the fusion of various ensemble combinations to mimic the WSI assessment by expert pathologists, to capture local and global context. Our results using a multi-scale approach demonstrate 3-9% improvement in classification accuracy when compared with models utilising a single magnification level. This advancement underscores the efficacy of attention-based models combined with multi-scale approaches in augmenting traditional assessment of WSIs. The implications of our findings are significant in enhancing glioma diagnosis and treatment planning in neuro-oncology, by enabling diagnostics in low-resource environments where molecular profiling is not available.

**Keywords:** glioma, multi-scale, deep learning, computational pathology

# 1 Introduction

Diffusely infiltrating gliomas are the most prevalent primary malignant adult brain tumors within the central nervous system (CNS) (Ostrom et al., 2021). The recent guidelines from World Health Organization (WHO) integrate histological features with molecular profiling for conclusive glioma diagnosis. These guidelines identify three major types pivotal for clinical stratification in gliomas: *oligodendroglioma* (IDH mutant and 1p/19q codeleted) (Grade 2, 3)); *astrocytoma* (IDH mutant (Grade 2, 3, 4)); and *glioblastoma*, (IDH wildtype (Grade 4)). Fig. 1 illustrates the simplified workflow of molecular features affecting the staging and subtyping of gliomas. These distinct tumor subtypes exhibit disparities in survival outcomes, clinical trial eligibility, and serve as crucial prognostic indicators guiding therapeutic decisions (Louis et al., 2016b; Pekmezci et al., 2017). Though visual evaluation of glass slides remains a gold standard for histologic assessment of adult infiltrating gliomas, it is not sufficient for their diagnostic categorization. Moreover, molecular analysis of tumors is not always feasible and can be time-consuming. Given the necessity of incorporating molecular information with histologic assessment to obtain a conclusive diagnosis, there is increasing evidence with advancements in deep learning (DL) based methods to directly predict the predominant molecular alterations from Hematoxylin and Eosin (H&E)-stained WSIs (Campanella et al., 2019; Louis et al., 2016a; Cifci et al., 2022; Innani et al., 2023).

The existing literature on analysis of gliomas based on WSI tends to concentrate on specific molecular alterations like IDH1/2, ATRX (Hewitt et al., 2023; Liu et al., 2020; Jin et al., 2021; Innani et al., 2023) or prognosis (Baheti et al., 2023a,b, 2024), and only a few studies have focused on the WHO 2021 glioma classification (Jin et al., 2021; Pei et al., 2021; Wang et al., 2023; Nasrallah et al., 2023). However, most of the WSI-based approaches have primarily developed DL models at a single magnification level (Lu et al., 2021; Baheti et al., 2023d), whereas tissue glass slides are digitized into WSI with pyramidal resolutions, enabling quantitative analysis at multiple scales. Unlike pathologists who assess tissue samples using a range of magnifications to capture detailed information at different levels, these prior studies often combine features early in their multi-scale approaches, potentially missing important context-specific details. Therefore, these methods fail to take advantage of the rich information a digitized WSI has to offer and fully leverage the tissue features across different fine-to-course resolutions from cellular to millimeter scale similar to the expert pathologist. Consequently, existing algorithms only extract limited information related to invasion, depth of tumors, and cell types, as they only assess a single magnification level. While methods for analyzing multi-scale WSI have been explored in various studies across different organs, including breast (Li et al., 2023), lung (Ding et al., 2023), kidney (Hou et al., 2022), and bowel (Deng et al., 2024), gliomas present unique challenges in this field due to their significant histologic heterogeneity (Sottoriva et al., 2013).

In this paper, we introduce a multi-scale late fusion approach aimed at capturing hierarchical diagnostic information across various magnification levels, such as cellular-scale (e.g., nucleus and micro-environment), tissue-scale (e.g., vessels and glands), and global-scale structures from heterogeneous pyramidal WSIs. Our objective is to achieve a more comprehensive slide-level classification of WHO 2021 glioma subtypes through a weakly supervised algorithm based on Multiple Instance Learning (MIL) (Maron and Lozano-Pérez, 1997). Our approach successfully identifies the WHO 2021 adult infiltrating gliomas classes

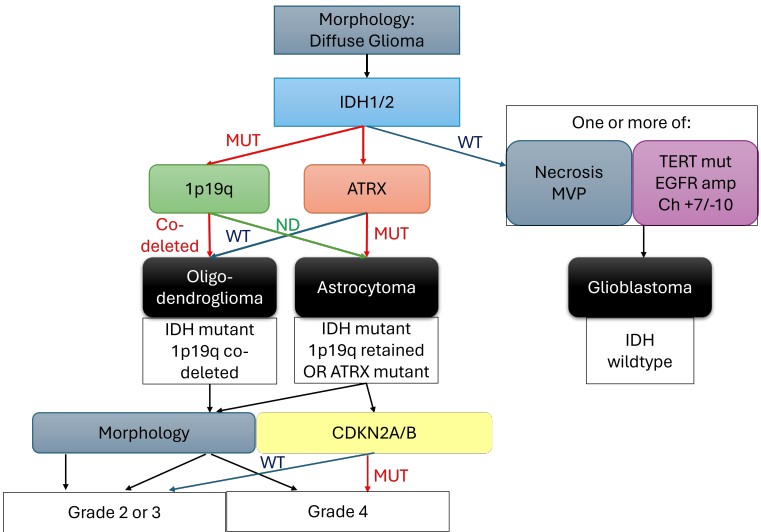

Figure 1: Simplified workflow for $5^{th}$ edition of WHO classification of CNS tumors for adult infiltrating gliomas. MVP - microvascular proliferation, WT - wildtype, MUT - mutant, ND - not deleted. The three major glioma classes considered in this study (oligodendroglioma, astrocytoma, and glioblastoma) are highlighted in black.

purely from patient WSI with a weakly supervised approach, thereby avoiding the annotation burden and potentially obviating the current need for molecular profiling.

## 2 Materials and Methods

### 2.1 Data

The evaluation of our proposed work is based on multi-institutional data from the TCGA Low-Grade Glioma Collection (TCGA-LGG)(Pedano et al., 2016) and The Cancer Genome Atlas Glioblastoma Multiforme (TCGA-GBM) (Scarpace et al., 2016) data collections, both publicly available via The Cancer Imaging Archive (TCIA) (Clark et al., 2013). To assign slide-level labels of glioma categories to WSI for our approach, reclassification information as per WHO 2021 was obtained from (Zakharova et al., 2022) and was verified with our board-certified neuropathologist. Our analysis focuses on a subset of these datasets, specifically targeting 654 out of 1,122 TCGA-GBM and TCGA-LGG patients for which both molecular alterations and formalin-fixed paraffin-embedded (FFPE) H&E-stained WSI are available, leading to a total of 1,320 H&E-stained WSIs either having 40x (mpp range: 0.2456 - 0.2533) or 20x (mpp range: 0.4993 - 0.504) maximum apparent magnification. To ensure consistency across analyses, 20x magnification was considered as the standard reference level, facilitating uniform processing and comparison in subsequent analyses. To rigorously assess model performance, we adopt a 10-fold cross-validation (CV) strategy partitioning the data into training (80%), validation (10%), and test (10%) sets. Importantly, this CV approach is conducted while stratifying at the patient level to account for cases where

multiple WSIs belong to the same patient. Data Distribution for each subgroup is presented in Table 1.

Table 1: Distribution of data as per WHO 2021 classification

| Class | WSIs (Patients) |
|---|---|
| Oligodendroglioma, IDH-mutant and 1p/19q-codeleted (grade 2,3) | 301 (141) |
| Astrocytoma, IDH-mutant (grade 2,3,4) | 403 (224) |
| Glioblastoma, IDH-wildtype (grade 4) | 616 (289) |

## 2.2 Approach

The overall pipeline of the proposed multi-scale architecture is presented in Fig. 2. Initially, each WSI of variable size is divided into non-overlapping patches (N) of 256 × 256 pixels at different scales corresponding to apparent magnification levels of 20x, 10x, 5x, 2.5x. At each magnification level, comprehensive patch-level curation is applied to retain the patches with informative tissue and discard patches with artifactual content (such as background, blurriness, pen markings, and dirt on the glass) by the approach proposed in (Baheti et al., 2023c). Each retained patch is encoded by an ImageNet pretrained ResNet-50 (He et al., 2016) to obtain a 1024-dimensional feature representation. We further employ a Multiple Instance Learning (MIL) approach using attention mechanisms (Ilse et al., 2018), specifically adapted for three-class glioma classification. Following the MIL assumption (Maron and Lozano-Pérez, 1997), where each patch within a WSI is treated as an instance and the WSI itself as a bag, our model processes WSIs represented as feature vectors of size N × 1024. Initially, each 1024 dimensional feature vector undergoes dimensionality reduction to size of 512, through a trainable fully connected (FC) layer. This reduced feature vector is then forwarded to the attention network, which consists of two parallel FC layers employing Tanh and Sigmoid (Dubey et al., 2022) activations respectively. The outputs of these layers each of size $N \times 256$, are elementwise multiplied to derive attention scores for each patch and passing through another linear layer resulting in an output of size $N \times 1$. A separate FC layer with softmax activation (Liu et al., 2016) is used to predict a class probability for each patch based on the 512-dimensional features, resulting in an output of size $N \times 3$. The attention scores ($N \times 1$) are used as weights to aggregate the patch-level predictions into a single prediction for the entire WSI, ensuring that more important patches have a greater influence on the final classification. Weighted loss is implemented to mitigate the class inbalance problem. The model is trained for 200 epochs with early stopping, Adam optimizer and dropout rate of 0.25. We introduce a late fusion paradigm designed to capture the inter-scale relationships employed by the pathologist while examining the tissue slide. Note that *late fusion* in this work is following the definition given by Lipkova et al. (2022), where it refers to the aggregation of individual model predictions to generate the final prediction. We perform experiments on individual magnification levels, as well as their different combinations, where separate models are trained for each scale, and are ensembled at the output stage by averaging the probabilities of each class predicted by individual models.

## 3 Results

Table 2 shows the performance results of the 10-fold CV on the test set at different magnification levels. The table includes metrics like Balanced Accuracy, Area Under the ROC

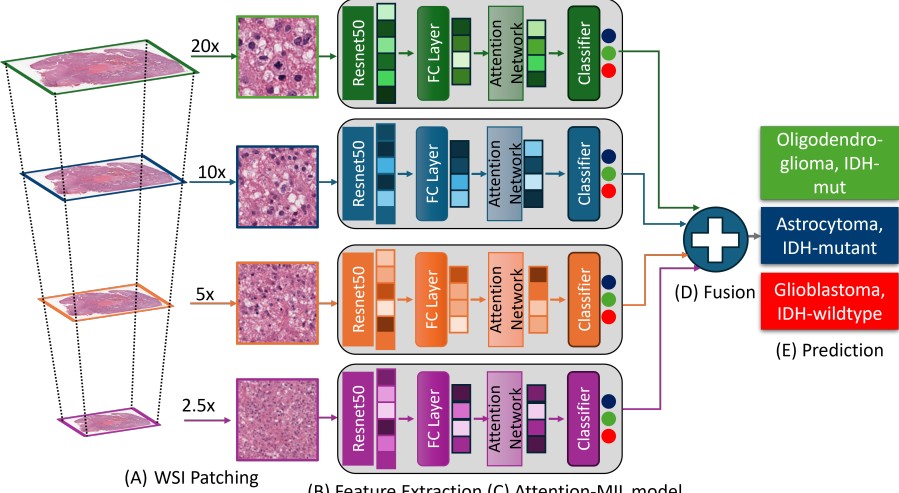

Figure 2: Schematic of our proposed multi-scale attention-MIL pipeline. (A) Tissue Segmentation followed non-overlapping patch extraction with curation. (B) Patches are encoded into features using ResNet-50. (C) Attention-MIL model is trained for each scale independently that outputs probabilities of each class. (D) Fusion is a late ensemble technique combining the results of each scale. (E) Final prediction of the underlying glioma type.

Curve (AUC), Sensitivity, and Specificity macro-averaged across 10-folds for various combinations of magnification level. These metrics collectively demonstrate the robust model performance in classifying the WHO 2021 subtypes WSI. Among the 10-CV results, the performance metrics vary across different magnifications, showcasing the model varying performance on different combination of scales. Notably, the results from combining all scales stand out as delivering the best performance among all the other combinations. Fig. 3 (A-C) represents separate ROC curves for each glioma subtype across 10-fold CV test cohorts. These plots are from the best-performing model, which is a fusion of all magnifications (20x, 10x, 5x, 2.5x). Fig. 3 D presents the confusion matrix of the 10-fold results.

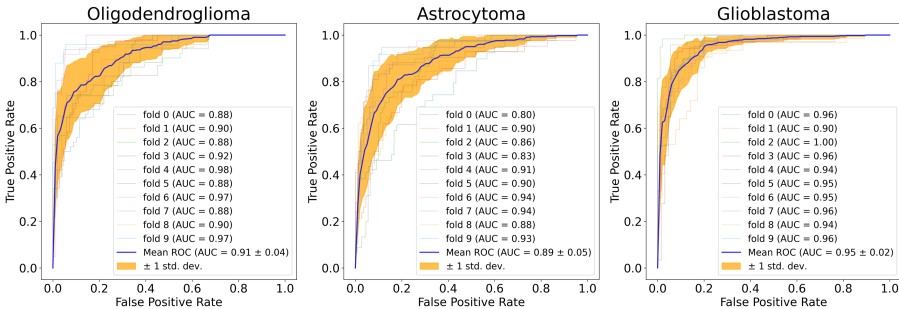

Figure 3: A-C. Receiver Operating Characteristic (ROC) Curves of the three classes.

Table 2: Multi-scale classification results averaged across 10-folds on test dataset in terms of Area Under Curve (AUC), Balanced Accuracy, Sensitivity and Specificity. Each row represents model trained on single or fusion of different magnifications and corresponding metrics with mean ± standard deviation. The ordered rank of performance is based on DELPHI-based recommendations for image analysis validation.

| Magnification | AUC | Bal. Acc. | Sensitvity | Specificity |
|---|---|---|---|---|
| 2.5x (o) | 0.8579 ± 0.0332 | 0.6876 ± 0.0598 | 0.6876 ± 0.0598 | 0.8488 ± 0.0261 |
| 5x (n) | 0.8793 ± 0.0327 | 0.7001 ± 0.0521 | 0.7001 ± 0.0521 | 0.86 ± 0.0295 |
| 2.5x_5x (m) | 0.8916 ± 0.0306 | 0.7334 ± 0.0495 | 0.7334 ± 0.0495 | 0.8744 ± 0.0277 |
| 20x (l) | 0.9023 ± 0.0325 | 0.7312 ± 0.0525 | 0.7312 ± 0.0525 | 0.8763 ± 0.0242 |
| 2.5x_10x (k) | 0.9034 ± 0.0267 | 0.7499 ± 0.0572 | 0.7499 ± 0.0572 | 0.8815 ± 0.0303 |
| 10x (j) | 0.8956 ± 0.0343 | 0.7585 ± 0.0656 | 0.7585 ± 0.0656 | 0.8844 ± 0.0333 |
| 2.5x_5x_10x (i) | 0.909 ± 0.0262 | 0.7544 ± 0.0623 | 0.7544 ± 0.0623 | 0.8851 ± 0.0348 |
| 5x_10x (h) | 0.9061 ± 0.0279 | 0.7556 ± 0.0611 | 0.7556 ± 0.0611 | 0.8861 ± 0.032 |
| 5x_20x (g) | 0.9103 ± 0.0328 | 0.7462 ± 0.0472 | 0.7462 ± 0.0472 | 0.8848 ± 0.023 |
| 10x_20x (f) | 0.9126 ± 0.0298 | 0.7576 ± 0.046 | 0.7576 ± 0.046 | 0.8894 ± 0.0212 |
| 2.5x_20x (e) | 0.907 ± 0.0288 | 0.7627 ± 0.029 | 0.7627 ± 0.029 | 0.8894 ± 0.0176 |
| 2.5x_5x_20x (d) | 0.913 ± 0.0295 | 0.7584 ± 0.0328 | 0.7584 ± 0.0328 | 0.8884 ± 0.0222 |
| 2.5x_10x_20x (c) | 0.9172 ± 0.0249 | 0.7684 ± 0.0508 | 0.7684 ± 0.0508 | 0.8927 ± 0.0254 |
| 5x_10x_20x (b) | 0.917 ± 0.0285 | 0.7681 ± 0.0492 | 0.7681 ± 0.0492 | 0.8937 ± 0.0239 |
| **2.5x_5x_10x_20x (a)** | **0.9185 ± 0.0254** | **0.7693 ± 0.0539** | **0.7693 ± 0.0539** | **0.8948 ± 0.0283** |

## 4 Discussion

In this study, we have demonstrated the effectiveness of integrating multi-scale features with MIL to directly identify the WHO 2021 classification of adult infiltrating diffuse gliomas from WSI. By leveraging information across different magnification levels, our approach enhances the diagnostic yield of WSIs, aiding clinicians with a robust approach for precise glioma diagnosis. The incorporation of multi-scale tissue analysis has shown superior performance compared to single-scale algorithms, highlighting the importance of capturing spatial context at various resolutions.

We sought a computational deep learning-based diagnostic workflow for infiltrating adult gliomas, utilizing H&E-stained WSI as the sole input. This approach addresses the insufficiency of visual inspection of glass slides alone in meeting the diagnostic criteria for tumor categorization set by WHO 2021. We utilized the reclassified TCGA-GBM and TCGA-LGG dataset and demonstrate the effectiveness of computational approaches for tumor classification as per WHO 2021 criteria. We further hypothesized that our model training and outcome is magnification-level dependent and accurate prediction is achieved by the ensemble of various magnifications to mimic the pathologist's behavior while examining WSIs. Pathologists usually use low magnification to identify features visible on a larger scale and to locate regions of interest for detailed examination at higher magnification. Our results demonstrates that the multi-scale approach achieves impressive performance with AUC scores being above 0.8 for each of the subtype of glioma for the best performing fusion model (the combination of 20x, 10x, 5x, 2.5x).

Our study stands out as one of the pioneering endeavors to systematically explore the permutation and combination of different magnification levels, offering a comprehensive evaluation of multi-scale features in glioma classification. The observed differences in AUC performance with regard to fusion of various ensemble combinations highlight the relation-

Table 3: Pairwise statistical comparison of models based on permutation testing across different magnifications. The first column lists the magnification combinations, and the "Ranking" column indicates their overall rank. The "Statistical Group" column groups models that are not statistically different from each other.

| Magnification | Ranking | Statistical Group | a | b | c | d | e | f | g | h | i | j | k | l | m | n | o |
|---|---|---|---|---|---|---|---|---|---|---|---|---|---|---|---|---|---|---|
| 2.5x_5x_10x_20x (a) | 1 | 1 | | 0.3303 | 0.1656 | 0.0586 | 0.0584 | 0.0301 | 0.0141 | 0.0044 | 0.0003 | 0.0028 | 0.0003 | 0.0003 | 0.0001 | 0 | 0 |
| 5x_10x_20x (b) | 2 | 1 | | | 0.3218 | 0.1614 | 0.1300 | 0.0354 | 0.0207 | 0.0035 | 0.0038 | 0.0050 | 0.0026 | 0.0006 | 0.0005 | 0 | 0 |
| 2.5x_10x_20x (c) | 3 | 1 | | | | 0.2105 | 0.1398 | 0.1171 | 0.0904 | 0.0359 | 0.0102 | 0.0103 | 0.0007 | 0.0012 | 0.0014 | 0 | 0 |
| 2.5x_5x_20x (d) | 4 | 1 | | | | | 0.4188 | 0.3541 | 0.2066 | 0.1392 | 0.0835 | 0.0607 | 0.0230 | 0.0066 | 0.0026 | 0 | 0 |
| 2.5x_20x (e) | 5 | 1 | | | | | | 0.4010 | 0.2791 | 0.1956 | 0.1393 | 0.0819 | 0.0325 | 0.0030 | 0.0134 | 0.0001 | 0 |
| 10x_20x (f) | 6 | 2 | | | | | | | 0.3453 | 0.2043 | 0.2004 | 0.0798 | 0.0525 | 0.0058 | 0.0299 | 0.0001 | 0 |
| 5x_20x (g) | 7 | 2 | | | | | | | | 0.3002 | 0.3001 | 0.1703 | 0.1210 | 0.0275 | 0.0394 | 0 | 0 |
| 5x_10x (h) | 8 | 2 | | | | | | | | | 0.4765 | 0.2531 | 0.2183 | 0.1094 | 0.0651 | 0 | 0 |
| 2.5x_5x_10x (i) | 9 | 2 | | | | | | | | | | 0.2728 | 0.1522 | 0.1115 | 0.0335 | 0 | 0 |
| 10x (j) | 10 | 2 | | | | | | | | | | | 0.4064 | 0.2287 | 0.1931 | 0.0028 | 0.0001 |
| 2.5x_10x (k) | 11 | 2 | | | | | | | | | | | | 0.2660 | 0.2127 | 0 | 0 |
| 20x (l) | 12 | 3 | | | | | | | | | | | | | 0.4899 | 0.0332 | 0.0010 |
| 2.5x_5x (m) | 13 | 3 | | | | | | | | | | | | | | 0.0003 | 0 |
| 5x (n) | 14 | 4 | | | | | | | | | | | | | | | 0.0896 |
| 2.5x (o) | 15 | 4 | | | | | | | | | | | | | | | |

ship between spatial resolution and diagnostic accuracy, underscoring the need for a subtle approach in leveraging multi-scale features for computational pathology tasks. The performance comparison across multiple magnification scales, as depicted in Table 2, reveals fascinating insights into the impact of input magnification on model efficacy. Notably, our findings indicate that the choice of magnification significantly influences model performance, with the highest levels of accuracy observed when all magnification levels are combined. Table 3 summarizes the statistical significance across all the considered multi-scale approaches, following the DELPHI-based recommendations for image analysis validation (Reinke et al., 2024; Maier-Hein et al., 2024), incorporating i) algorithmic ranking, and ii) statistical significance testing. For this analysis we divided the test data into multiple non-overlapping subsets, ensuring balanced class representation. We then computed an average rank for each of the subsets across all multi-scale approaches, and aggregated these average rankings to produce a conclusive overall ranking. All approaches were then placed in a ranked order and their average rankings were randomly permuted (i.e., 100,000 permutations), in a pair-wise manner. Corresponding pairwise p-values shown in Table 3 were computed to determine the pair-wise statistical significance and report actual differences between the ordered ranked approaches. These p-values are reported in an upper triangular matrix (Table 3) revealing the statistical insignificance of the first five approaches ($p > 0.05$), and hence clustered together as 'group 1'. This group is significantly better than the sixth approach ($p = 0.0301$), indicated by a vertical and horizontal line. We note that the top-performing approaches of group 1 (a, b, c, d, & e) involve the combination of multiple magnification levels, while the last group (n & o) captures information in coarse magnification levels.

We also observe that the result obtained at intermediate magnification level (5x, 10x) is comparable to the ensemble of all magnifications. One plausible interpretation of this observation is that low magnification levels offer a broader field of view, providing enhanced architectural information and facilitating comprehensive tissue sampling. Conversely, high power levels have increased cytologic detail capturing key diagnostic features but with a narrower field of view and intermediate magnification levels may strike a balance, capturing both low-power and high-power information effectively. Subsequently, our hypothesis of designing models to mimic the pathologist's behavior of examining tissue at low magnification and subsequently focusing on regions of interest at higher magnification levels leads to more robust performance.

While our study demonstrates promising advancements in predicting WHO 2021 classification subtypes directly from WSI utilizing an Attention-MIL approach with late ensemble technique, several limitations should be acknowledged. Despite efforts to mitigate class imbalance after reclassification, inherent disparities in subtype distribution may persist, potentially biasing the model towards more prevalent classes. Furthermore, the reliance on relatively small patch sizes of $256 \times 256$ pixels might restrict the model's capacity to capture broader spatial context and architectural patterns present in larger tissue regions within WSI. A solution to this approach could be offered by leveraging a WSI based approach such as Streaming CLAM (Dooper et al., 2023). Addressing these limitations is crucial for optimizing the applicability and accuracy of our model in real-world clinical scenarios. Moving forward, several avenues present themselves for further enhancing the robustness and applicability of our model. Firstly, the evaluation of our model on multi-site datasets is paramount to assess the generalizability across diverse patient populations and acquisition systems. Additionally, we aim to assess the interpretability of the obtained decisions by analysing attention scores as heatmaps derived from MIL model. Furthermore, extending similar methods to other tasks in oncology, such as tumor grading and prognostic prediction, could provide valuable insights into the decision-making processes of advanced AI systems in clinical practice. Lastly, there exist opportunities for refining our methodology to leverage the latest advancements in self-supervised learning models, particularly those based on Vision Transformers (ViTs)(Chen et al., 2024). Unlike the present approach, which relies on feature embeddings generated from pre-trained ImageNet models, incorporating ViTs pretrained on large scale WSIs can potentially yield contextually rich representations, thereby improving the model's discriminative power. Complementary information from different modalities like MRI and clinical data could also be leveraged by our model to gain a more holistic understanding of the tumor, thereby enhancing its diagnostic capabilities. Lastly, by generating heatmaps from the attention scores, model interpretability can be enhanced (Baheti et al., 2023c). By pursuing these future directions, we aim to further advance the state-of-the-art in computational pathology and contribute towards more accurate and reliable diagnostic tools for precision oncology.

This work contributes to advancing the field of computational pathology and also holds promise for improving clinical decision-making and patient outcomes in the realm of glioma management, ultimately paving the way towards improved healthcare outcomes and enhanced patient care.

## Acknowledgments and Disclosure of Funding

Research reported in this publication was partly supported by the Informatics Technology for Cancer Research (ITCR) program of the National Cancer Institute (NCI) of the National Institutes of Health (NIH), under award number U01CA242871 and Lilly Endowment, Inc. through its support for the Indiana University Pervasive Technology Institute. The content of this publication is solely the responsibility of the authors and does not represent the official views of the NIH or any other funding body.

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
