# OpenReview forum: "Multi-scale Whole Slide Image Assessment Improves Deep Learning based WHO 2021 Glioma Classification"
_MICCAI.org/2024/Workshop/COMPAYL — COMPAYL 2024_

### Official Review · Reviewer_Cms6 · 2024-07-09
**Review of Submission 14**

**Custom Rating:** 3
**Confidence:** 5

**Review:**

Summary:

The manuscript proposes a computational pipeline for glioma classification from H&E whole slide images. The method extracts non-overlapping patches, encodes them using a pre-trained ResNet50, and employs an attention-based multiple instance learning (MIL) approach to classify individual patches. Patch predictions are then aggregated, with weights based on the attention scores. Independent classifiers are trained at four magnification levels (20x, 10x, 5x, 2.5x), and various ensemble combinations are evaluated. The authors find that the best performing ensemble combines all four magnification levels.


Pros:
- The paper is well-motivated, explaining the relevance of the task and providing guidelines to identify glioma types based on histological and molecular features.
- The study demonstrates the effectiveness of integrating multi-scale features with a MIL approach for diffuse glioma classification.
- The results emphasize the importance of incorporating spatial context at different resolutions for WSI classification.

Cons:
- The technical novelty is limited. Recent advancements include visual transformers pre-trained on histopathology data for richer and more specific embeddings than those obtained with ImageNet-pretrained CNNs. Additionally, more advanced attention mechanisms could be explored for the aggregation stage.
- The paper lacks details on how accuracy, specificity, and sensitivity are computed (micro or macro averaging). Furthermore, detailed per-class classification metrics are not presented.
- The manuscript acknowledges class imbalance in the discussion but does not specify any strategies employed to mitigate its effects. It would also be valuable to compare the subtype prevalence in the dataset to real-world distributions.
- The approach lacks interpretability. Utilizing the MIL framework and attention scores, the authors could have generated heatmaps depicting patch importance.

---

### Official Review · Reviewer_KTSV · 2024-07-10
**The study is well-motivated and well-designed, and the manuscript is well-written. However, there are some parts of the methodology that can be improved.**

**Custom Rating:** 3
**Confidence:** 5

**Review:**

Pathologists assess tissue samples using a range of magnifications to capture detailed information at different levels. In this study, the authors developed a multi-scale approach to glioma classification to mimic the pathologists' workflow by training individual models at four magnification levels (20x, 10x, 5x, 2.5x) and applying late fusion.

Comments regarding the application:
- In many parts, it is mentioned that ResNet50 pre-trained on ImageNet was used to perform the feature extraction of the patches, which is an out-of-domain model. It should be better to use a feature extractor of a histology foundation model. However, in the discussion part, it is stated that the UNI histology foundation model will be a future implementation to perform the feature extraction of the patches.
- The late fusion is described as a weighted average probability. It could be argued that this method might not be interpreted as late fusion since fusion should be included in training.
- It would be better to use balanced accuracy instead of accuracy since the dataset is imbalanced. Additionally, a 10-cv was applied, and the average performance was reported, but the standard deviation should also have been reported.
- In the Discussion, it is stated that the findings indicate that the choice of magnification significantly influences the model performance, but no statistical comparison is performed. Thus, this statement is not well-argued, and no clear assumptions could be made about how significant is the magnification level for a model. In addition, it is mentioned that despite efforts to mitigate class imbalance after reclassification, inherent disparities in subtype distribution may persist, potentially biasing the model towards more prevalent classes. However, methods to mitigate the class imbalance could be performed, such as class weight adjustment, to mitigate the class imbalance problem. Lastly, it is stated that the evaluation of our model on multisite datasets is paramount to assess the generalizability across diverse patient populations and acquisition systems. However, if an encoder from a histology foundation model had been used to perform the feature extraction from the patches, it would have helped mitigate generalizability issues.

---

### Official Review · Reviewer_RVd3 · 2024-07-15

**Custom Rating:** 2
**Confidence:** 4

**Review:**

**Summary**

This paper presents a multi-scale approach for the classification of adult infiltrative gliomas from HE whole-slide images. The methods consists in 1) embedding non-overlapping (256, 256) patches extracted at 4 different resolutions (0.5, 1, 2 & 4 microns per pixels) using a ResNet50 pretrained on ImageNet 2) training 4 resolution-specific attention-based multiple instance learning models 3) averaging each model prediction in a late fusion step. The authors demonstrate that combining features from multiple magnification levels improves classification accuracy by 3-9% compared to single magnification models.

**Strengths And Weaknesses**

Strength:
- the motivation for the work is well explained.
- the results demonstrates the feasibility of accurately classification glioma from HE whole-slide images.
- the authors conducted thorough experiments to appropriately identify which magnification levels combination yields the best performance.

Weakness:
- the authors do not provide any statistical significance tests of the results presented in table 2.
- the authors do not provide standard deviation for the results presented in table 2.
- the authors do not provide enough details on some of the key aspects of the method (how do they train the patch-level classifier when only slide-level labels are available? it's not clear either how they go from (N, 256) feature vectors into (N, 1) attention scores as element wise multiplication of (N, 256) feature vectors doesn't result in a (N, 1) vector).
- certain aspects of the method (using ImageNet features, training each resolution-specific model separately, not exploring any fusion techniques other than simple averaging) are quite limiting when compared with previously published papers that also leverage multi-scale input (e.g. *Deep Multi-Magnification Networks for multi-class breast cancer image segmentation*, DJ Ho et al., 2021)
- unsure if this work fits well within the scope(s) of the COMPAYL workshop.

---

### Decision · Program_Chairs · 2024-07-16

Accept